# Seroprevalence and associated risk factors of brucellosis, Rift Valley fever and Q fever among settled and mobile agro-pastoralist communities and their livestock in Chad

**Ranya Özcelik**[1☯]*, **Mahamat Fayiz Abakar**[2☯]*, **Michel Jacques Counotte**[3], **Fatima Abdelrazak Zakaria**[2], **Pidou Kimala**[2], **Ramadane Issa**[2], **Salome Dürr**[1]

**1** Veterinary Public Health Institute, Vetsuisse Faculty, University of Bern, Bern, Switzerland, **2** Institut de Recherche en Elevage pour le Développement, N'Djamena, Chad, **3** Institute for Social and Preventive Medicine, University of Bern, Bern, Switzerland

☯ These authors contributed equally to this work.
* ranya.oezcelik@unibe.ch (RO); fayizalhilou@gmail.com (MFA)

**Citation:** Özcelik R, Abakar MF, Counotte MJ, Abdelrazak Zakaria F, Kimala P, Issa R, et al. (2023) Seroprevalence and associated risk factors of brucellosis, Rift Valley fever and Q fever among settled and mobile agro-pastoralist communities and their livestock in Chad. PLoS Negl Trop Dis 17(6): e0011395. https://doi.org/10.1371/journal.pntd.0011395

## Abstract

Brucellosis, Rift Valley fever (RVF) and Q fever are zoonoses prevalent in many developing countries, causing a high burden on human and animal health. Only a few studies are available on these among agro-pastoralist communities and their livestock in Chad. The objective of our study was to estimate brucellosis, RVF and Q fever seroprevalence among Chadian agro-pastoralist communities and their livestock, and to investigate risk factors for seropositivity. We conducted a multi-stage cross-sectional serological survey in two rural health districts, Yao and Danamadji (966 human and 1041 livestock (cattle, sheep, goat and equine) samples)). The true seroprevalence were calculated applying a Bayesian framework to adjust for imperfect diagnostic test characteristics and accounting for clustering in the study design. Risk factors for each of the zoonotic diseases were estimated using mixed effects logistic regression models. The overall prevalence for brucellosis, Q fever and RVF combined for both regions was estimated at 0.2% [95% credibility Interval: 0–1.1], 49.1% [% CI: 38.9–58.8] and 28.1% [%CI: 23.4–33.3] in humans, and 0.3% [%CI: 0–1.5], 12.8% [%CI: 9.7–16.4] and 10.2% [%CI: 7.6–13.4] in animals. Risk factors correlating significantly with the respective disease seropositivity were sex for human brucellosis, sex and Q fever co-infection for animal brucellosis, age for human Q fever, species and brucellosis co-infection for animal Q fever, age and herd-level animal RVF seroprevalence within the same cluster for human RVF, and cluster-level human RVF seroprevalence within the same cluster for animal RVF. In Danamadji and Yao, Q fever and RVF are notably seroprevalent among agro-pastoralist human and animal communities, while brucellosis appears to have a low prevalence. Correlation between the seroprevalence between humans and animals living in the same communities was detected for RVF, highlighting the interlinkage of human and animal transmissible diseases and of their health, highlighting the importance of a One Health approach.

**Data Availability Statement:** All relevant data are within the manuscript and its Supporting information files and analysis scripts.

**Funding:** This study was funded by the Swiss Federal Food Safety and Veterinary Office (https://www.blv.admin.ch/blv/en/home.html) by the grant number 1.17.o accquired by the authors SD and MFA and the Wolfermann-Nägeli Foundation (https://www.fundraiso.ch/en/sponsor/wolfermann-naegeli-stiftung) grant number 2016/19 accquired by the authors SD and MFA. The funders had no role in study design, data collection and analysis, decision to publish, or preparation of the manuscript.

**Competing interests:** The authors declare that no competing interests exist for the present study.

## Author summary

Infectious diseases transmitted between humans and animals, called zoonotic diseases, pose a global threat to human and animal health. Furthermore, diseased animals, especially livestock, can compromise the financial resources and livelihood of their owners as these depend on healthy animals for milk or meat production, or for agricultural work purposes. Brucellosis, Q fever and Rift Valley fever are two bacterial and one viral zoonotic disease that were found to be prevalent among many human-animal communities living in close contact, such as it is the case among Chadian agro-pastoralists. Limited data are available on the current status of these diseases in Chad. In this study, the authors investigated the prevalence of these three diseases among humans and their livestock (cattle, sheep, goats, horses and donkeys) by collecting blood samples and conducting serological analyses in two rural regions of Chad, Danamadji and Yao. Results point towards high Q fever and Rift Valley fever seroprevalences (13–49% and 10–28%, respectively), and low prevalence of brucellosis (< 1%), and towards a positive association between human and animal Rift Valley fever seroprevalence. With these findings, the study hopes to support current and future zoonotic disease surveillance and control efforts within the regions.

## 1. Introduction

Zoonotic diseases represent a major global threat to public health and the global economy. The World Health Organization (WHO) estimates that around one billion illnesses and millions of human deaths occur every year globally from zoonotic diseases [1]. More than 60% of the emerging infectious diseases reported globally are of zoonotic origin, and around 75% of the latest 30 emerging human pathogens have originated in animal populations [2]. In Africa, many zoonotic diseases are endemic or reemerge in regular outbreaks in human and animal populations [3,4]. Three highly relevant zoonotic diseases are brucellosis, Rift Valley fever (RVF) and Q fever, that cause a huge burden in humans and animals [5–9]. For many decades, the presence, circulation and periodic outbreaks of these three diseases have cost human lives, caused chronic diseases and the disrupted livelihoods of people by affecting the health of their animals, in particular in communities where human nutrition and economic wellbeing depend largely on animal husbandry.

Brucellosis is caused by *Brucella spp*. and is widely spread in Sub-Saharan Africa, with a major economic and health impact on livestock production and health, as well as on human health and livelihood [5,10–12]. It is transmitted from infected animals to humans while handling body fluids, sick animals and aborted material, or through the consumption of unprocessed animal products, such as unpasteurized milk and fresh meat [13–16]. In humans, brucellosis presents with a wide range of symptoms including intermittent fever, arthralgia, myalgia, abortion, fatigue, and in some cases neurologic disorders with a drastic reduction in quality of life [17,18]. It can cause systemic infections involving multiple organs and organ systems, yet can remain unrecognized due to the lack of pathognomonic symptoms [14]. Relapses, chronic localized infections in multiple organs, and delayed convalescence exists, with consequences such as orchitis or epididymitis, or a rare, yet mostly lethal endocarditis [16,19,20]. In animals, brucellosis is known to reduce fertility, decline in milk production, and cause abortion in multiple livestock, such as cattle, camels and goats [21–23]. In some cases, animals can be asymptomatic carriers of *Brucella spp*. and develop symptoms in the later chronic phase, such as intermittent fever, joint hygroma and orchiepididymitis [24,25].

Rift Valley fever is a vector-borne zoonotic viral disease and since its discovery in 1930, outbreaks have occurred and serological evidence was found among humans and animals in various African countries, such as Chad, Kenya, Nigeria and Mauritania, to name a few [26–32]. The disease is commonly found in domesticated and wild ruminants, such as buffaloes, goats, sheep, cattle, camels and equine [31,33–35]. Rift Valley fever is transmitted amongst livestock by infected adult female mosquitos, especially after heavy persistent rains with flooding when the infected eggs hatch. Transmission to humans from livestock occurs by direct contact with infected blood, tissue, aborted material, or birthing fluid of animals, or by bites of infected mosquitos [36,37]. RVF virus infections in humans can remain asymptomatic in early stages, yet about 8% of them result in a severe disease, manifested by hemorrhagic fever, hepatitis, or encephalitis with a mortality rate of up to 50% [38]. In animals, RVF is responsible for high mortality rates in young animals during new outbreaks, caused by acute viremia and hepatitis [39]. Systemic infection by RVF virus in animals manifests in the liver, the eyes, and, when present for a longer period, also in the central nervous system, leading to high and undulant fever, hepatitis, neurological disorders, ocular dysfunction, and abortion [34,40].

Q fever is caused by the bacteria *Coxiella burnetii* and has been found in varying levels of endemicity among human and animal populations on the African continent, such as The Gambia, South Africa and Chad [15,32,41,42]. The disease is transmitted to humans through inhalation of contaminated aerosols discharged by fluids during birthing or abortion, urine and feces of infected animals, or by the consumption of unprocessed animal products such as raw milk [43–46]. Lambing season in sheep combined with favoring wind conditions can cause large outbreaks of Q fever cases in humans [47]. Q fever in humans cause unspecific febrile illness, however about 40% of the cases develop pneumonia and hepatitis [48]. Up to 30–52% of severe Q fever cases lead to chronic fatigue syndrome that can persist for multiple years [44]. In animals, *Coxiella burnetii* is known to cause multi-focal acute to chronic infections, manifesting in endocarditis, pneumonia, and abortion [44].

While there are numerous studies on human and animal seroprevalence of these three zoonotic diseases in many sub-Saharan African countries, such as Kenya [49–51], Uganda [52,53], and Ethiopia [21,54,55], there are only a few reporting on the diseases' seroprevalence in Chad. Between 1999 and 2000, Schelling et al (2003) revealed human brucellosis and Q fever seroprevalence of 3.8% and 1% in the Chadian provinces Chari-Baguirmi and Kanem, respectively [15]. The same study found seroprevalence for brucellosis and Q fever, of 7% and 4% in cattle, 0% and 13% in goats, 0% and 11% in sheep, and 0.4% and 80% in camels, respectively. Abakar et al (2014) estimated the cattle RVF, Q fever and brucellosis seroprevalence on the southeastern shore of Lake Chad in 2014 at 37.8%, 7.8%, and 5.7%, respectively [32]. Nevertheless, both studies were conducted several years ago, were carried out for a certain limited geographical region within Chad, or focused on animals or human seroprevalence only.

In Chad, the rural population lives mostly within agro-pastoralists lifestyle settings. Pastoralism is an extensive farming method based on the exploitation of natural vegetation and is often dependent on mobility of farmers and their livestock, and is the dominant economic activity in dry and semi-arid regions within the Sahelian-belt of the African continent, including Chad [56]. Due to their remoteness to public infrastructure in rural settlements and their constant mobility, agro-pastoralist communities are frequently at disadvantage regarding health care and veterinary service access' [57]. Despite humans and animals living in close contact within agro-pastoralist communities [58], evidence of the correlation between human and animal seroprevalence of relevant zoonoses is so far lacking. Investigating such correlation could provide evidence for quantifying the relevance of transmission between humans and their livestock. To construct future zoonotic disease surveillance systems and control efforts integrating human and animal health conjointly, updated and multispecies seroprevalence

studies are necessary to assess the level of endemically prevalent or reoccurring zoonotic diseases.

The present study aimed at quantifying the seroprevalence of brucellosis, RVF, and Q fever in humans and their livestock (cattle, sheep, goats, and equine (horse and donkey)) in two rural health districts Danamadji and Yao, in Chad. We further investigated individual and community-level factors, which influence the seroprevalence of these diseases. Knowledge derived from this study serves as updated epidemiological data and can be used as a baseline for implementing integrated human and animal One Health surveillance systems and for better understanding risk factors associated with serological zoonotic disease presence, especially in the context of rural African settings. Furthermore, this study is the first to our knowledge to report on brucellosis, Q fever and RVF seroprevalence in equine kept as livestock in Chadian communities and in the region.

## 2. Material and methods

### 2.1. Ethics statement

The study has been submitted to and approved by the Ethics Committee of Northwest and Central Switzerland (EKNZ) (project id 2017–00884) and by the Comité National de Bioéthique du Tchad (CNBT) (project id 134/PR/MESRS/CNBT/2018). Formal written consent was obtained from study participants and from those whose animals were sampled after introducing our study to each visited community and before data collection took place.

### 2.2. Study regions

The study was conducted among animal and human populations in two rural health districts in Chad, Yao and Danamadji. Both regions have been intervention zones of the Support Project for the Health Districts in Chad (PADS), a development project funded by the Swiss Development and Cooperation Department and managed by the Swiss Tropical and Public Health Institute [59]. Yao health district is located in the Lake Fitri Basin in the Batha province in the Sahel and covers an estimated 141'217 inhabitants. Danamadji health district is located in the middle of the Chari River valley in the sub-humid zone in the Moyen-Chari province, bordering the Central African Republic and covers 123'788 inhabitants. Together, the two districts cover 31 functional health zones, defined as geographic areas of responsibility of a health facility [60]. According to the last general animal population census, the total number of animal population in Chad was estimated at 93'803'192 livestock and 34'638609 poultry farming. The Province of Batha contains the largest portion of the livestock population at national level with around 12.6%. Meanwhile, the livestock population in Moyen-Chari Province represents only 1% of national livestock [61].

Yao and Danamadji health districts have been selected as our study regions due to existing synergies between our project and PADS, and the infrastructure (roads, housing opportunity for the research team, health centers for storing samples cool) and resources (knowledgeable local community members supporting access to communities, up-to-date maps of known camps) available.

### 2.3. Study design, target population and data collection

The cross-sectional study took place in January and February 2018. A list of existing camps and villages in the two regions was available from PADS. Camps in the context of Chadian demographics are temporary settlements of mostly mobile communities consisting of fairly easily mountable and demountable tents and animal pens. Villages are settlements of sedentary

communities consisting of mostly clay, and sometimes brick and cement houses. Upon arrival at a study site the chief of the community was approached first. This usually came along with a community gathering at a central place within the camp or the village, as a result of the study team's arrival. The study purpose was verbally explained in Arabic or in the local language. In most cases the chief of the community solely decided to allow the study to be conducted, yet on some occasions he (in our study always men) conferred the decision with the elderly assembly.

A multi-stage cluster sampling was conducted with villages and camps selected as first-level clusters, and humans and animals within the selected clusters as second level. Selection of the first level clusters was conducted proportional to human population size, retrieved from available demographic data. The sample size was calculated using R statistical software [62]. Due to the unknown prevalence of brucellosis, Q fever and RVF within these regions, we assumed a design prevalence of 50% for humans and animals, delivering the highest sample size. The precision was set at 0.1 for a 95% confidence interval to calculate the sample size for a simple random sampling process ($n_{random}$). The sample size for two-stage cluster sampling ($n_{clusters}$) was based on $n_{random}$, the ICC (intra-cluster correlation coefficient) and the number of humans and animals sampled per cluster (m) according to Eq (1) [63,64]:

$$n_{cluster} = n_{random}(1 + ICC(m - 1)) \tag{1}$$

We set m at 20 individuals to ensure a logistically optimized number of clusters to be sampled. We assumed an ICC of 0.2 [50]. This led to the targeted sampling size of 24 clusters (12 camps and 12 villages) in each region and 20 humans and 20 livestock within each cluster to be sampled. We included women and men older than 14 years of age and excluded pregnant women. No obviously ill people were sampled. We included all livestock found at the study site that was above two years of age according to the information provided by the owner. Cattle, goats, sheep, horses and donkeys were sampled. Obviously ill, hence often isolated animals, were excluded from sampling.

In humans, blood was collected from the median cubital vein by nurses, while for livestock, blood was collected by veterinarians or veterinary technicians from the jugular vein, both in dry serum collection tubes (5ml). Samples were rested until the end of each sampling day and blood clots were removed to prevent the serum from becoming hemolytic. Thereafter serum samples were stored at 4°C in a fridge of the closest local health center in each health district. After the return to the central laboratory at the Institut de Recherche en Elevage pour le Développement (IRED) in N'Djamena, the capital of Chad, samples were centrifuged and transferred to cryotubes before being frozen at -20°C until diagnostic analysis.

## 2.4. Laboratory analyses

A Competitive Enzyme-Linked Immunosorbent Assay (C- ELISA) was applied to detect immunoglobulin G (IgG) antibodies against RVF in human and animal samples (ID Screen Rift Valley fever Competition Multi-species, Grabels, France). Rose Bengal test was used to detect the presence of antibodies against brucellosis in human and animal samples. Q fever IgG antibodies in animal samples were detected using the ID Screen Q Fever Indirect Multi-species kits. Human samples were analyzed for Q fever using Panbio Coxiella burnetii (Q fever) IgG ELISA (STANDARD DIAGNOSTICS INC. the Republic of Korea (www.alere. com)). The manufacturers' protocols and the cut-offs indicated to distinguish between positive and negative samples were used without any modification during sample analysis. All samples were tested in duplicate except for Q fever because of the limited amount of reagents available.

## 2.5. Statistical analysis for the estimation of the seroprevalence

Data analysis was performed using R software version 4.0.1 (2020-06-06) [65]. We first calculated the apparent seroprevalence for each disease by dividing the number of positive samples by the total sample size per region and species. Second, we estimated the true seroprevalence by adjusting the apparent seroprevalence for imperfect test sensitivity and specificity, and for the cluster design by applying a Bayesian framework as described by the HOTLINE Project [66]. Further methodological steps for the estimation of the true prevalence are described in the S1 Methods, S1 and S2 R Scripts, together with the original data (S1 Data).

## 2.6. Risk factors for brucellosis, Q fever and Rift Valley fever seropositivity in humans and animals

Information on potential risk factors for brucellosis, Q fever and RVF seropositivity were collected during sampling by interviews with the study participants and animal owners. Univariable and multivariable mixed effect logistic regression (MELR) models were applied to each disease separately, with individual seropositivity (no = 0, yes = 1) of humans or animals being the outcome variable. The hierarchical structure of the models includes *cluster within region* as a random effect for those models including explanatory variables on individual level only. For models including explanatory variables on cluster level, *region* only was included as a random effect.

First, univariable MELR models using the following fixed effect variables for the models in humans were conducted: animal apparent seroprevalence (of brucellosis, Q fever or RVF, respectively) within the same cluster (continuous variable), sex (female vs male), age (years of age as a continuous variable), sampling site (camp vs village), and co-infection with one of the other two diseases (present vs not present). The following fixed effect variables were selected for the animal models: human apparent seroprevalence (of brucellosis, Q fever or RVF, respectively) within the same cluster (continuous variable), sex (female vs male), age (binary, $<3$ vs $\geq 3$, (in years)), species (cattle, small ruminants (sheep and goats), equine (horses and donkeys)), sampling site (camp vs village), and co-infection with one of the other two diseases (present vs not present). Second, fixed effect variables associated with a p-value $< 0.2$ in the univariate MELR models were selected for building multivariable MELR models of the same hierarchical structure as in the univariable models. We visually checked for multicollinearity among explanatory variables by using the correlation plots produced by the ggpairs function from the GGally package [67] (S1 and S2 Figs). The final multivariable MELR models were identified by a stepwise backwards selection of the explanatory variables and by choosing the model with the lowest Akaike's Information Criterion (AIC) as the selection criteria [68]. Variables with coefficient p-values of $< 0.05$ were considered statistically significant.

## 3. Results

### 3.1 Demographics of the sampled population

In total, we collected 966 human and 1041 animal blood samples (388 from bovines, 155 goats, 369 sheep, 82 horses and 47 donkeys). The median age of the sampled humans in both regions combined was 35 years of age and ranged between 14 to 100 (Inter quartile range: 25–48 years). The majority of humans sampled in both regions combined were men with 66.4% (n = 641) of the samples, while women were sampled less with 33.6% (n = 325). Cattle, sheep and goats were almost equal parts either younger than three years of age ($n_{young\_cattle} = 206$, 53.1%; $n_{young\_sheep} = 203$, 55.0%; $n_{young\_goat} = 75$, 48.4%), or three years of age or older, while in horses and donkeys the majority ($n_{horse\_old} = 79$, 96.3% and $n_{donkey\_old} = 39$, 83.0%) were

three years of age or older (S1 Table). In bovine, sheep and goats more females were sampled ($n_{bovine\_female}$ = 250, 64.4%; $n_{sheep\_female}$ = 269, 72.9%; $n_{goat\_female}$ = 118, 76.1%), while among horses and donkeys more males ($n_{horse\_male}$ = 61, 74.4% and $n_{donkey\_male}$ = 31, 66%) were sampled (S2 Table).

## 3.2 Seroprevalence

Of 966 human samples, 959 were tested for brucellosis, 960 for Q fever and 954 for RVF. All 1041 animal samples were tested for brucellosis, 975 for Q fever and 1002 for RVF. An overview of the number of humans and animals tested, the number of diagnostically positive samples, apparent seroprevalence (AP) and the true seroprevalence (TP) per species and region for all three diseases were provided in S3 Table. In Danamadji only 12 goats and seven equines were sampled, which was too low for interpretable estimations on the seroprevalence, hence they are subsequently not presented in the further results, nor discussed.

In humans, when both regions combined, Q fever TP (TP: 49.1%, [95% Credibility Interval: 38.9–58.8%]) was the highest out of the three zoonotic diseases, followed by RVF (28.1%, [% CI: 23.4–33.3%]), and finally brucellosis (0.2%, [%CI: 0–1.1%]) (Fig 1, S3 Table). Similarly, in animals (all species combined), the overall TP was highest for Q fever (12.8%, [%CI: 9.7–16.4%), followed by RVF (10.2%, [%CI: 7.6–13.4%]), and finally brucellosis (0.3%, [%CI: 0–1.5%]).

In humans, brucellosis TP was estimated similarly low in both regions, with 0.2% [%CI: 0–1.2%] in Danamadji, 0.5% [%CI: 0–3.1%] in Yao (Fig 1). Similarly, brucellosis TP in all animal species combined was overall low in both regions (Yao: 0.8%, [%CI: 0–4.1%]; Danamadji: 0.5%, [%CI: 0–2.9%]). However, when stratified by species and region, animals in Yao had higher brucellosis TP than those in Danamadji, while the highest brucellosis TP was observed among sheep (4.7%, [%CI: 0–14.0%]) in Yao.

When stratified by region, human Q fever TP was estimated almost twice as high in Danamadji (63.0%, [%CI: 52.3–74.7]), compared to Yao (35.1%, [%CI: 22.3–46.7]) (Fig 1). On the contrary, animal Q fever TP (all species), was about 3% higher in Yao compared to Danamadji (Yao: 14.5%, [%CI: 10.1–19.7]; Danamadji: 11.4%, [%CI: 7.6–15.9]). When stratified by species and region, sheep in Yao had the highest (21.1%, [%CI: 12.5–32.8]), and equine in Yao had the lowest (4.2%, [%CI: 0–11.7]) Q fever TP of all animal species. Overall, human Q fever TP was significantly higher than in animals revealed from non-overlap of the 95% credibility intervals.

Contrary to human Q fever TP results, human RVF TP was estimated higher in Yao (32.9%, [%CI: 25.1–39.7]) compared to Danamadji (25.9%, [%CI: 19.9–31.7]), whereas all animal RVF TP was higher in Danamadji (13.6%, [%CI: 9.6–18.2]) compared to Yao (7.9%, [%CI: 4.8–11.5]) (Fig 1). When stratified by species and region, sheep in Danamadji had the highest RVF seroprevalence (18.6%, [%CI: 11.8–26.2]).

## 3.3 Risk factors for brucellosis in humans and animals

The univariable MELR models revealed that human brucellosis seropositivity is positively yet not significantly correlated with the animal brucellosis apparent seroprevalence of the same cluster (S4 Table). Being RVF seropositive is negatively yet not significantly correlated with humans being brucellosis seropositive. Women have lower, yet statistically not significant odds of being brucellosis seropositive compared to men. Living in a village, being Q fever co-infected, and age did not show any association with brucellosis seropositivity. No multivariable MELR was built for risk factors of human brucellosis because none of the p-values in the univariable analysis was below the threshold (p-value < 0.2), except sex (Odds Ratio [OR]: 0.3,

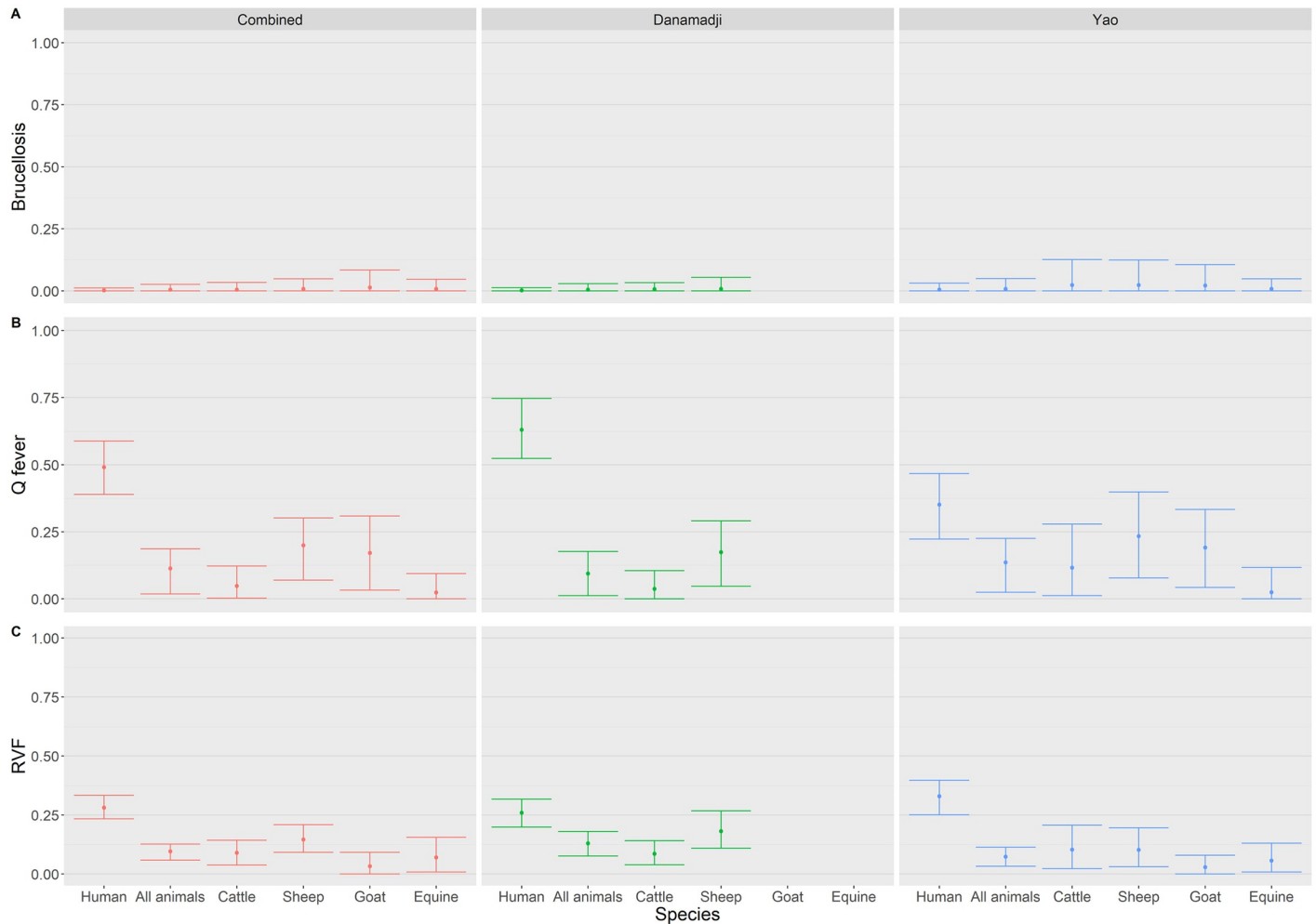

**Fig 1. Human and animal brucellosis, Q fever and Rift Valley fever seroprevalences.** Estimations on human and animal brucellosis (**A**), Q fever (**B**) and RVF (**C**) true seroprevalence (middle point) and their credibility intervals (outliers) estimated using a Bayesian framework adjusting for clustering and imperfect test characteristics, in Yao and Danamaji, and both regions combined.

95% Confidence Interval [95%CI]:0.1;1.5, p-value: 0.135), which resulted in the best-fitted model for human brucellosis (Table 1).

In the univariable analysis, animal brucellosis seropositivity was positively yet not significantly correlated with the human brucellosis apparent seroprevalence within the same cluster (S5 Table). A positive association was also detected for age (being three years or older), being Q fever co-infected (compared to not being infected) and for small ruminant species (compared to cattle). Being female (compared to male) was found to be statistically significantly correlated with brucellosis seropositivity. Equine species (compared to cattle) and living in a village (compared to living in a camp) are negatively yet not significantly correlated with animal brucellosis seropositivity. The variables species (OR = 1.6, 95%CI = 0.8;3.1, p-value: 0.186), being Q fever co-infected (OR = 1.6, 95%CI = 0.8;3.3, p-value: 0.165) and female sex (OR = 2.4, 95%CI = 1.2;4.7, p-value: 0.009) were further selected as variables for the multivariable analysis, because of their p-value < 0.2. In the final multivariable mixed effects regression model for animal brucellosis, females (OR = 2.3, 95%CI = 1.1;4.9, p-value: 0.0253) and having

**Table 1. Risk factors for individual level animal and human brucellosis, Q fever and Rift Valley fever seropositivity, of conjoint human and animal communities from Yao and Danamadji, Chad.**

| Disease | Species | Variable | OR | 95% CI | p-value |
|---|---|---|---|---|---|
| **Brucellosis** | Humans | Sex | | | |
| | | Male (ref. level) | - | - | - |
| | | Female | 0.3 | 0.1;1.5 | 0.135 |
| | Animals | Sex | | | |
| | | Male (ref. level) | - | - | - |
| | | Female | 2.3 | 1.1;4.9 | 0.025 |
| | | Q fever co-infection: | - | - | - |
| | | absent (ref. level) | | | |
| | | present | 2.0 | 1.0;4.0 | 0.049 |
| **Q fever** | Humans | Age (step 1 year) | 0.99 | 0.98;1.00 | 0.018 |
| | Animals | Species | | | |
| | | Cattle (ref. level) | - | - | - |
| | | Equine | 0.7 | 0.3;1.6 | 0.340 |
| | | Small ruminants | 2.313 | 1.5;3.6 | <0.001 |
| | | Brucellosis co-infection: | | | |
| | | absent (ref. level) | - | - | - |
| | | present | 1.9 | 0.9;3.8 | 0.076 |
| **Rift Valley fever** | Humans | RVF apparent seroprevalence in animals (step 1%) | 4.28 | 1.4;13.6 | 0.0134 |
| | | Age (step 1 year) | 1.02 | 1.01;1.03 | <0.001 |
| | Animals | RVF apparent seroprevalence in humans (step 1%) | 12.9 | 2.8;58.7 | <0.001 |

The best fitting model is presented, either univariable or multivariable, depending on model selection criteria. Model outcomes of risk factors are presented as odds ratios (OR), 95% confidence intervals (95% CI) and p-values (p-value).

a Q fever coinfection (OR = 2.0, 95%CI = 1.003;3.982, p-value: 0.0491) were significantly associated with animals being brucellosis seropositive (Table 1).

## 3.4 Risk factors for Q fever in humans and animals

The univariable MELR models for humans revealed that Q fever seropositivity is positively yet not significantly correlated with the animal Q fever apparent seroprevalence within the same cluster, living in a village, being female, and a brucellosis co-infection present (S6 Table). Having a RVF co-infection present was not associated with human Q fever seropositivity. On the other hand, an increase in age was negatively and significantly correlated with human Q fever seropositivity (OR = 0.99, 95%CI = 0.98;1.0, p-value: 0.0178). No multivariable MELR was built because none of the p-values in the univariable analysis showed a p-value < 0.2, except age, which resulted in the best-fitted model for human Q fever (Table 1).

In animals, the univariable MELR models revealed that small ruminants (compared to cattle), brucellosis seropositive animals, RVF seropositive animals and females are positively correlated with Q fever seropositivity, however only small ruminants (OR = 2.2, 95%CI = 1.4;3.6, p-value 0.000549), female sex (OR = 1.8, 95%CI = 1.2;2.8, p-value: 0.00641) and brucellosis seropositivity (OR = 1.7, 95%CI = 0.9;3.4, p-value: 0.125) were below the threshold of p-value < 0.2 (S7 Table). Age did not show an association with Q fever seropositivity. Human Q fever apparent seroprevalence within the same cluster, equine species (compared to cattle) and animals living in a village are negatively correlated with animal Q fever seropositivity, although not statistically significant. The best fitting multivariable MELR model shows that small

ruminant species is a significant risk factor for animal Q fever seropositivity (Table 1), while having a brucellosis co-infection present and equine species are non-significant risk and protective factors of animal Q fever seropositivity, respectively.

### 3.5 Risk factors for Rift Valley fever in humans and animals

The univariable MELR models revealed that human RVF seropositivity is positively yet not significantly correlated with living in a village (S8 Table). Animal RVF apparent seroprevalence within the same cluster (OR = 4.0, 95%CI = 1.3;12.3, p-value: 0.0137) and an increase of age (OR = 1.02, 95%CI = 1.01;1.03, p-value: 2.64e-05) are positively and significantly correlated with human RVF seropositivity. Having a brucellosis co-infection present and female gender are negatively, however not significantly, correlated with RVF seropositivity. Being Q fever seropositive neither increases nor reduces the odds of RVF seropositivity. The best fitting multivariable model revealed that an increase in age of one year and a percent increase in RVF seropositivity of animals within the same cluster are significant risk factors for higher odds of human RVF seropositivity (Table 1).

The univariable MELR models revealed that animal RVF seropositivity is positively and significantly correlated with human RVF apparent seroprevalence within the same cluster (12.0, 95%CI = 2.9;52.7, p-value: 0.00069) (S9 Table). Animal Q fever co-infection, living in a village, animals of three years of age or older and small ruminants (compared to cattle) are correlated with higher, however not significant odds of animal RVF seropositivity, of which only age (1.4, 95%CI = 0.9;2.2, p-value: 0.101) was selected for the multivariable model. Sex neither increases nor decreases the odds of animal RVF seropositivity. Animal brucellosis seropositivity is significantly and negatively correlated with animal RVF seropositivity (0.1, 95%CI = 0.0;1.0, p-value: 0.0443). Equine species (compared to cattle) are negatively yet not significantly correlated with animal RVF seropositivity. The initial multivariable model was then built using brucellosis co-infection, human RVF apparent seroprevalence and age as explanatory factors. The final multivariable model for animal RVF seropositivity included only one explanatory variable and revealed that a percent increase of human RVF apparent seroprevalence within the same cluster is a significant risk factor for animal RVF seropositivity (12.9, 95%CI = 2.8;58.7, p-value: <0.001).

## 4. Discussion

In this study we present the seroprevalence of brucellosis, RVF and Q fever among rural settled and mobile agro-pastoralist communities and their livestock in two regions in Chad. We further present risk factors associated with individual human and animal seropositivity, including the association of diseases between the human and animal populations at the village level. Three relevant zoonotic diseases are studied simultaneously in humans and their livestock in poorly accessible, rural communities, including equine species, which have thus far rarely been included in investigations for these zoonotic diseases. In line with previous research, our study confirms the presence of Q fever and RVF, and the low brucellosis seroprevalence in rural Chad [15,32].

Our study reported very low brucellosis seroprevalence in humans and animals in both regions with value less than 1%. This suggests that brucellosis could be regarded as an insignificant concern regarding the health of humans and their livestock. Brucellosis is mostly considered endemic in the African continent, however with varying yet rather low levels of seroprevalence overall [69–71]. Previous estimations of human and animal brucellosis seroprevalence in other regions of Chad were higher compared to our findings. The differences in estimated seroprevalence between studies could be due to the different timing of data

collection, such as a seasonal effect, or due to the low sensitivity of the Rose-Bengal test under the conditions of chronically diseased individuals [72,73]. Schelling et al. (2003) estimated 4% seroprevalence in cattle [15], while Abakar et al. (2014) reported 5.7% seroprevalence in cattle with the Rose-Bengal test and 11.9% with ELISA [32]. These estimations are based on the apparent seroprevalence without correction for imperfect tests [32], or the TP calculated using the Rogan and Gladen correction [15], which is known to be not ideal for very low prevalence [74]. The estimation of the TP hence differs by prevalence estimation methods of choice. This is confirmed by our results showing an unneglectable difference between the apparent and true seroprevalence estimations for low-prevalent brucellosis (S3 Table).

The risk factor analysis for human brucellosis seroprevalence within our study revealed that women have significantly lower odds of brucellosis seropositivity. A previous study investigating demographic factors related to human brucellosis seroprevalence among agro-pastoralist communities in Uganda found that gender was not a significant factor for brucellosis [52]. A study conducted in rural Tanzania reported that human brucellosis was associated with assisted parturition during the abortion in cattle, sheep or goats [75]. In pastoral settings, obstetric tasks in livestock are carried out more frequently by men [15], while female household heads are likelier to adhere to safer practices because of more frequent exposure to health workers while attending antenatal care or child welfare clinics [76]. This might be the cause for a higher rate of brucellosis transmission to men rather than women, thus potentially explaining the significant difference in the relation of gender observed in our study.

Our results suggest that female animals have higher odds of being brucellosis seropositive. A higher odds of animal seropositivity for brucellosis among females might be due to their prolonged life expectancy compared to males, as they are kept alive longer due to breeding and milk production reasons [21,55]. Although we are here not able to investigate this hypothesis by our data because we only used a binary age categorization, our argumentation is in line with a previous study conducted in the Lake Chad region found similar results regarding female cattle being more likely brucellosis seropositive, and attributed this to a possible age effect [32]. Female livestock, those providing milk especially, such as cattle, goats and sheep, tend to be kept in the herd for a longer time for producing offspring and milk, hence risks of disease transmission, such as brucellosis, could accumulate over time, compared to males that have a shorter life expectancy as they are most likely to be slaughtered for their meat. Individual animal level brucellosis seropositivity was also significantly higher in Q fever positive animals, and vice-versa. The co-infection with brucellosis and Q fever in animals (in the respective studied ruminants), as well as in humans, is a previously documented finding within a study conducted amongst pastoralist communities and their livestock in Sokoto State, Nigeria [77] and ruminants in Guinea [78]. The presence of a certain disease might negatively impact the overall immune system and hence allow for further diseases to manifest in the respective host [79].

Human Q fever seroprevalence was estimated at almost 50% in our study. In contrast to our findings, Schelling et al (2003) estimated a lower human Q fever seroprevalence of 3.8% in the southeast shore of Lake Chad region Lake Chad region [15]. In animals, we found a seroprevalence for Q fever of 12.8% in all animal species together, which is comparable with a previous study from Egypt [80]. In the study of Schelling et al (2003), Q fever apparent seroprevalence in cattle was estimated at 7.8% (95% CI: 5.6–10.1) [15]. In a study in cattle in Kenya, Q fever seropositivity was estimated lower at 5.7% (95% CI: 2.1%- 9.4%), while for goats and sheep estimates were higher at 18.2% (95% CI: 13.7%- 22.7%), and 13.0% (95% CI: 6.4%- 19.6%), respectively [81]. Our findings show a similar pattern when disaggregated by species, where cattle (and equine) have lower Q fever seroprevalences than goats and sheep.

Human Q fever infections are known to result mainly from animal reservoirs, while cattle and small ruminants are shown to be the key spreaders, and lambing and calving key events for transmission to humans [80,82–84]. Furthermore, Q fever seroprevalence can be traced back to certain environmental factors. Q fever in cattle has previously been associated with drinking from the watercourse and well water [85], while human infections were associated with close contact with water points that livestock had access to [82]. Chadian livestock owners take their livestock regularly to visit water points, as we can confirm from our field study observations, and this is also where herds from different camps and villages gather [86]. Such encounters might facilitate continuous contamination of the water, as well as enable inter-herd and inter-species disease transmission. Besides the epidemiological effect of water points, human cases have also been associated with Q fever spores being transported by wind [83,87]. The environmental and climatic settings in our study sites constitute the more dry land (Yao) and the more humid land (Danamadji), with both of the regions going through rainy and dry seasons. This might affect Q fever seroprevalence in humans and animals accordingly. While the estimated seroprevalence in animals overall was found to be comparable between the two regions, human Q fever seroprevalence was about twice as high in Danamadji (63.0%) compared to Yao (35.1%), and the same effect was observed for cattle.

Data in our study suggests that the odds of human Q fever seropositivity are significantly lower with increasing age. Although the association was found to be statistically significant, the effect size was, with an OR of 0.99, negligible. Previous studies conducted in Ethiopia and the Gambia reported a not significant yet positive correlation between increasing age and human Q fever seropositivity [41,55], highlighting the unexpected negative effect of age on Q fever seropositivity in our study.

The risk factor analysis for Q fever seropositivity in animals revealed that small ruminants, compared to cattle, have significantly higher odds of being seropositive. These findings are consistent with previously reported results from Kenya [81]. Differences in the odds of Q fever seropositivity according to animal species can be explained by deviating susceptibility of the respective hosts and of its immune response [88], or by differing management practices of and veterinary care resources for cattle and small ruminants. More attention is typically attributed to cattle herd and health management compared to small ruminants, and especially during calving and lambing, as in agro-pastoralist settings cattle tend to be considered as an investment of higher value than small ruminants [89]. Furthermore, small ruminants, especially in a low number as 20 per household, are considered as part of rather poor households [90], which tend to have low economic resources for veterinary care. These aspects could potentially have led to an increased risk of transmission within, and lacking health care management of small ruminants. While previous Q fever seroprevalence studies conducted in Chad have found Q fever seropositive dromedaries, another livestock very extensively kept by predominantly mobile pastoralists, as high as 73% of herds [57], in our study we did not come across dromedary herds to be able to compare such data.

We estimated overall human RVF seroprevalence at 28.1% and overall animal seroprevalence at 10.2%. While human RVF seroprevalence was higher in Yao, compared to Danamadji, the seroprevalence was similar between regions among cattle, and for sheep, higher RVF seroprevalence was found in Danamadji. In a study conducted in Kenya, both cattle (1.4% (95% CI 0.5–2.22)) and human (0.5%, 95% CI 0.2–0.8) seroprevalence were found to be considerably lower [91]. In another study conducted in Chad, Abakar et al (2014) estimated higher RVF apparent seroprevalence in the Lake Chad region in cattle (37.8%), but similar estimates in goats (18.8%) and sheep (10.8%) compared to our findings [32]. Mosquitos are considered one of the most important drivers of RVF infections in humans and animals [92–94]. Complex environmental factors, such as the El Nino Southern Oscillation, influence ocean

temperatures, rainfall and land temperature, which in return steers the density of mosquito populations [95–97]. Such environmental factors in RVF endemic regions might be a reason for the difference in disease seroprevalence between regions and countries. It is also important to note that RVF is endemic in the Sahelian region of Africa, which is not the case in east Africa where the disease is mostly associated with floods and heavy rains. This was particularly the case during the recent RVF outbreak that occurred in 2016 on the Niger side of the Lake Chad region, which resulted in several human cases including more than 30 cases of death [98].

The risk factor analysis within our study revealed that for increasing apparent seroprevalence of RVF within animals of the same village or camp, humans have higher odds of being seropositive. Vice-versa, animals are 13 times more likely to be seropositive with increasing human seroprevalence within the same village or camp. This shows the zoonotic nature of RVF. Our findings are in line with a previous study on human and animal RVF from Uganda, where human RVF seropositivity was found to be significantly associated with animal RVF seropositivity [53]. Animal to human direct transmission is associated with certain animal handling practices, such as contact with aborted material or sick animals and consumption of unprocessed animal products [37,57]. These practices are particularly common among communities living in close proximity with and dependent on their livestock for daily tasks such as milking, slaughtering, and transport of goods and people, as it is the case in rural Chad. We also identified increasing age as a risk factor for humans being RVF seropositive. Previous studies are in line with these findings [99], which can be explained by a cumulative and hence increasing risk for having experienced RVF transmission with growing age.

Goat and equine were less often kept by agro-pastoralist communities in villages and camps in the Danamadji region, compared to in the Yao region. Because of the very low number of sampled goats and equine in Danamadji, the sample size for this region's analysis is too low to make meaningful comparisons to the same animal species from the Yao region. In future studies that aim for regional comparison between certain animal species, special attention should be given to adapt the sampling strategy accordingly.

In our study, the estimations were based on the serological status deriving from the immune response of a past infection, rather than the direct detection of the pathogens or a current episode of a clinical case. This provides a rough estimation of the presence of respective diseases for the time period of the life of sampled individuals until the sampling time point, yet not a comprehensive view on real-time disease presence. The three zoonoses of concern in our study have however complex vector-environment-host components to their life cycle (such as; persistence in vectors and environment, reintroduction by human and animal movement and by changing climate conditions). Consequently, the nature of such pathogen life cycles can either enable the endemic persistence, or the reoccurring outbreak of such infections within respective human and animal populations. Thus, such baseline seroprevalence studies, even if conducted by a simple cross-sectional design, can be of guidance for informing current and future surveillance, as well as outbreak control, and epidemic or pandemic preparedness efforts. Worth to be mentioned are also the impact of recent outbreaks of the respective diseases within the studies region, which can lead to higher seroprevalence results yielded among sampled individuals and animals, compared to if no recent outbreaks would have occurred. This might have especially been the case for the 2018 onwards occurring RVF outbreaks reported from eastern African countries (Kenya, Rwanda, South Sudan, Sudan, and Uganda), partially bordering Chad [100].

Within the vision to strengthen public and animal health services in rural Chad, we aimed at using the outcome of the here presented seroprevalence study, combined with the outcomes of a retrospective cross-sectional survey on community-based observations of clinical signs in

humans and their livestock, to inform the development of a near real-time community-based One Health surveillance system that is currently in its trial phase in the two same regions, Danamadji and Yao.

## Conclusion

In this study we estimated the human, cattle, sheep, goat and equine true seroprevalence of three endemic zoonoses, brucellosis, Q fever and RVF, among agro-pastoralist communities and their livestock in two rural areas in Chad. While Q fever and RVF can be considered moderately to highly prevalent zoonotic diseases, brucellosis was found to be of low relevance, with seroprevalence below 1%, in both humans and animals. In the case of RVF, we were able to show a positive association between the seropositivity in human and animal cases, highlighting the interlinkage of human and animal transmissible diseases and their health, respectively. Although brucellosis and Q fever are relevant zoonoses as well, we could not detect this association in our data.

Conducting such baseline seroprevalence studies to provide a benchmark on prevalent zoonotic diseases can be used to inform the public, animal and environmental health authorities on the importance of intersectoral collaboration to better address the burden of zoonoses in the respective region or county. In Chad, the outcome of this study can be used for guiding future surveillance programs and interventions aiming at enhancing the health of humans, and their livestock, with an integrative One Health approach. This consequently enables the provision of better health services tailored to the needs of current zoonotic disease presence in the respective areas. Our findings might not only be important to respective health authorities and practitioners in Chad, but also to neighboring countries in the Sahel due to the high mobility of nomadic pastoralists and their livestock, as well as live animal and livestock product trade across country borders, driving the spread of various infectious diseases globally.

## Supporting information

**S1 Table. Number and proportion of animals sampled from each age category, separated by species.** NA stands for a missing observation.
(DOCX)

**S2 Table. Number (count) and proportion of animals sampled by species and sex.** NA stands for a missing observation.
(DOCX)

**S3 Table. Human and animal disease seroprevalence (%).** Number of samples tested (Tested), number of seropositive samples (Positive), apparent seroprevalence (AP), and true seroprevalence using Bayesian modeling adjusting for imperfect test characteristics and clustering (TP). Data presented for the regions Danamadji and Yao, as well as combined for both regions.
(DOCX)

**S4 Table. Univariable analysis results of risk factors tested for human brucellosis seropositivity in Yao and Danamadji, Chad.**
(DOCX)

**S5 Table. Univariable analysis results risk factors tested for animal brucellosis seropositivity in Yao and Danamadji, Chad.**
(DOCX)

**S6 Table. Univariable analysis results risk factors tested for human Q fever seropositivity in Yao and Danamadji, Chad.**
(DOCX)

**S7 Table. Univariable analysis results is factors tested for animal Q fever seropositivity in Yao and Danamadji, Chad.**
(DOCX)

**S8 Table. Univariable analysis results risk factors tested for human RVF seropositivity in Yao and Danamadji, Chad.**
(DOCX)

**S9 Table. Risk factors tested for animal RVF seropositivity in Yao and Danamadji, Chad.**
(DOCX)

**S1 Fig. Visualization of the correlation of potential risk factor variables from Yao and Danamadji, Chad, for which in the univariable multi regression model of human data revealed a p-value < 0.2.** These variables were then implemented in the respective multivariable models. No strong correlation was found for any combination, except for Age (count) and Age (category), which were never used in the same models. The age is presented in years and the animal RVF apparent seroprevalence in %.
(TIF)

**S2 Fig. Visualization of the correlation of potential risk factor variables from Yao and Danamadji, Chad, for which in the univariable multi regression model of animal data revealed a p value < 0.2.** These variables were then implemented in the respective multivariable models. No strong correlation was found for any combination, except for Age (count) and Age (category), which were never used in the same models. The age is presented in years and the human RVF apparent seroprevalence in %.
(PNG)

**S1 Methods. Description of the methodology estimating the true prevalence using a Bayesian framework.**
(DOCX)

**S1 R Script. R statistical software script 1.**
(PDF)

**S2 R Script. R statistical software script 2.**
(PDF)

**S1 Data. Original data.**
(ZIP)

## Acknowledgments

The authors would like to thank the study participants and local human and animal health authorities in the two study areas. We particularly would like to express our gratitude to the local communities' members who accepted to participate in the survey and the blood sample collection. Furthermore, we would like to thank the data collection team in the field and the technicians involved in the samples analysis process in the laboratory. The authors acknowledge the PADS (Programme d'appui aux districts sanitaires au Tchad) for collaboration and synergies.

## Author Contributions

**Conceptualization:** Mahamat Fayiz Abakar, Salome Dürr.

**Data curation:** Ranya Özcelik, Michel Jacques Counotte, Fatima Abdelrazak Zakaria, Pidou Kimala, Ramadane Issa.

**Formal analysis:** Ranya Özcelik, Michel Jacques Counotte, Fatima Abdelrazak Zakaria, Pidou Kimala, Ramadane Issa.

**Funding acquisition:** Salome Dürr.

**Investigation:** Ranya Özcelik, Ramadane Issa.

**Methodology:** Ranya Özcelik, Mahamat Fayiz Abakar, Michel Jacques Counotte, Fatima Abdelrazak Zakaria, Pidou Kimala, Ramadane Issa, Salome Dürr.

**Project administration:** Ranya Özcelik, Mahamat Fayiz Abakar, Salome Dürr.

**Resources:** Ranya Özcelik, Mahamat Fayiz Abakar, Salome Dürr.

**Software:** Ranya Özcelik, Michel Jacques Counotte.

**Supervision:** Mahamat Fayiz Abakar, Salome Dürr.

**Validation:** Ranya Özcelik, Salome Dürr.

**Visualization:** Ranya Özcelik, Michel Jacques Counotte.

**Writing – original draft:** Ranya Özcelik.

**Writing – review & editing:** Mahamat Fayiz Abakar, Michel Jacques Counotte, Fatima Abdelrazak Zakaria, Pidou Kimala, Ramadane Issa, Salome Dürr.

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
