## [Decision Letter · Decision Letter 0]

20 Sep 2022

Dear Dr. Oezcelik,

Thank you very much for submitting your manuscript "Seroprevalence and associated risk factors of brucellosis, Rift Valley fever and Q fever among settled and mobile agro-pastoralist communities and their livestock in Chad" for consideration at PLOS Neglected Tropical Diseases. As with all papers reviewed by the journal, your manuscript was reviewed by members of the editorial board and by several independent reviewers. The reviewers appreciated the attention to an important topic. Based on the reviews, we are likely to accept this manuscript for publication, providing that you modify the manuscript according to the review recommendations. 

Sincerely,

Ernesto T. A. Marques, M.D./Ph.D

Academic Editor

Esther Schnettler

Section Editor

Reviewer's Responses to Questions

**Key Review Criteria Required for Acceptance?**

**Methods**

-Are the objectives of the study clearly articulated with a clear testable hypothesis stated?

-Is the study design appropriate to address the stated objectives?

-Is the population clearly described and appropriate for the hypothesis being tested?

-Is the sample size sufficient to ensure adequate power to address the hypothesis being tested?

-Were correct statistical analysis used to support conclusions?

-Are there concerns about ethical or regulatory requirements being met?

Reviewer #1: The authors do not include an explanation for how their cut-offo of seropositive vs. seronegative samples for each disease. Please include this information

Reviewer #2: (No Response)

Reviewer #3: the objectives were clearly articulated but the selection of the two districts must be justified. the population was not clearly described (the number of inhabitants and the total number of animals per site should be mentioned in the Materials and Methods section). the sampling process must be more detailed

**Results**

-Does the analysis presented match the analysis plan?

-Are the results clearly and completely presented?

-Are the figures (Tables, Images) of sufficient quality for clarity?

Reviewer #1: The figures could be improved by:

-Including the average value for the estimation on true seroprevalance in Fig 2 & Supporting Fig 2.

-Providing a clear explanation of the units for each panel in Supporting Figs 1 & 2

Reviewer #2: (No Response)

Reviewer #3: the results are clearly and completely presented 

tables can be grouped into a single

figure 2 is unreadable

**Conclusions**

-Are the conclusions supported by the data presented?

-Are the limitations of analysis clearly described?

-Do the authors discuss how these data can be helpful to advance our understanding of the topic under study?

-Is public health relevance addressed?

Reviewer #1: - Starting at Line 396, the authors mention that the "higher odds of animal seropositive for brucellosis among females might be due to their prolonged life expectancy compared to males due to them being kep alive longer" however they do not explicitly state how their data supports this observation.

- The authors do not discuss when the most recent outbreak that occured for each of these diseases. This could significantly contribute to the seroprevelance.

Reviewer #2: (No Response)

Reviewer #3: the conclusion were supported by the data présented 

the limitations were not clearly described 

the authors discuss how these data can be helpful to advance our understanding of the topic under study

**Editorial and Data Presentation Modifications?**

Reviewer #1: Rift Valley fever virus was write with "Fever" capitalized, however it should be lower-case.

There is no hyphan between agro-pastoralists. It should just be "agropastoralists"

Line 21: zoonoss should be "zoonoses"

Line 22: Add "a" between only and few

Line 23: Remove "three"

Lines 48-55 would be best if it were truncated and left to the discussion/conclusions section

Line 72: Should read "aborted"

Line 94: Please include relevant references explaining the high mortality rates in young animals

Line 98: should read "burnetti"

Line 164-166: Can be removed since it isn't relevant to what was included in the study.

167: Remove word "being"

Line 262: lower-case "human"

Line 377: remove "of"

Line 386: "Significantly"

Line 448: remove "an"

Line 467: "differences"

Line 473: "humans"

Line 478: "aborted"

Line 481-482: Please clarify what you mean by "as is also happns in rural Chad"

Line 490-491 does not make sense

Reviewer #2: (No Response)

Reviewer #3: (No Response)

**Summary and General Comments**

Reviewer #1: This is a well written manuscript of a cross-sectional study that identify seroprevalence of brucellosis, Q fever, and Rift Valley fever virus (RVFV) in humans and livestock within two districts of Chad. This important study aimed to identify risk factors associated with seropositivity for each of these diseases, paired seropositivity, and it was the first study include equine in analyses of this kind. Correlations between RVFV-seropositive livestock and humans was identified, highlighting the zoonotic nature of RVFV. Similar correlations were not observed for brucellosis and Q fever. It is important to continue to perform these types of studies to provide data that can aid in surveillance and determining future health practices within these regions. Only minor changes are requested that I believe will improve the quality and readability of this manuscript.

Reviewer #2: This study on RVF, Q fever and brucellosis has been well conducted and presented. It respond to a major issue of zoonosis among rural pastoralist population and their livestock in Chad. the approach used if new and very clever as it combine both human and animals including horses and donkeys, which are not usually studied. Except for minor corrections, the authors should improve the English language quality

Reviewer #3: -This is an epidemiological and serological study of three abortive diseases in humans and animals in two districts of Chad, which are important for public health. One of the strengths was to focus on human and animals but several points deserve to be detailed specially in the materials and methods section. 

-Sex should be replaced with gender for human in the manuscript 

-Author must justify the choice of 20 human for the sampling process 

-Investigating the serological status of humans and animals towards the three diseases without having any idea about the vaccination status of the study population could bias all results

-The author should explain why all samples were not tested for the three diseases

PLOS authors have the option to publish the peer review history of their article (what does this mean?). If published, this will include your full peer review and any attached files.

Reviewer #1: No

Reviewer #2: No

Reviewer #3: No

Figure Files:

Data Requirements:

Reproducibility:

References

---

## [Decision Letter · Decision Letter 1]

22 May 2023

Dear Dr. Oezcelik,

We are pleased to inform you that your manuscript 'Seroprevalence and associated risk factors of brucellosis, Rift Valley fever and Q fever among settled and mobile agro-pastoralist communities and their livestock in Chad' has been provisionally accepted for publication in PLOS Neglected Tropical Diseases.

Best regards,

Ernesto T. A. Marques, M.D./Ph.D

Academic Editor

Esther Schnettler

Section Editor

The reviewers still concerned about the writing of the article that can be improved by a more rigorous grammatical review. No technical issues appear to remain, so please review the grammar one more time before your final submission.

Reviewer's Responses to Questions

**Key Review Criteria Required for Acceptance?**

**Methods**

-Are the objectives of the study clearly articulated with a clear testable hypothesis stated?

-Is the study design appropriate to address the stated objectives?

-Is the population clearly described and appropriate for the hypothesis being tested?

-Is the sample size sufficient to ensure adequate power to address the hypothesis being tested?

-Were correct statistical analysis used to support conclusions?

-Are there concerns about ethical or regulatory requirements being met?

Reviewer #1: (No Response)

Reviewer #2: The author names: remove “and” between the first two author names

Double check the English grammar and spelling across the document

Study regions: Use the correct English spelling for numbers : 141’217; 123’788; 93’803’192; 34’638609 : inverted comma should be replaced by comma

Table 1: highlight (bold) the statistically significant p-value

Reviewer #3: Major revision

**Results**

-Does the analysis presented match the analysis plan?

-Are the results clearly and completely presented?

-Are the figures (Tables, Images) of sufficient quality for clarity?

Reviewer #1: (No Response)

Reviewer #2: (No Response)

Reviewer #3: (No Response)

**Conclusions**

-Are the conclusions supported by the data presented?

-Are the limitations of analysis clearly described?

-Do the authors discuss how these data can be helpful to advance our understanding of the topic under study?

-Is public health relevance addressed?

Reviewer #1: (No Response)

Reviewer #2: (No Response)

Reviewer #3: (No Response)

**Editorial and Data Presentation Modifications?**

Reviewer #1: (No Response)

Reviewer #2: (No Response)

Reviewer #3: (No Response)

**Summary and General Comments**

Reviewer #1: (No Response)

Reviewer #2: The authors have improved the overall quality of thier manuscript which is now much more understandable. However, they should improve the english quality. Sadly they have remove their previous figure 1 on study site localisation. This figure was important as not everyone is familiar with Chad.

Reviewer #3: (No Response)

PLOS authors have the option to publish the peer review history of their article (what does this mean?). If published, this will include your full peer review and any attached files.

Reviewer #1: No

Reviewer #2: No

Reviewer #3: No

---

## [Editor Report · Acceptance letter]

20 Jun 2023

Dear Dr. Özcelik,

We are delighted to inform you that your manuscript, "Seroprevalence and associated risk factors of brucellosis, Rift Valley fever and Q fever among settled and mobile agro-pastoralist communities and their livestock in Chad," has been formally accepted for publication in PLOS Neglected Tropical Diseases.

Best regards,

Shaden Kamhawi

co-Editor-in-Chief

Paul Brindley

co-Editor-in-Chief
